

# Validation of Terrestrial Water Storage Variations as Simulated by Different Global Numerical Models with GRACE Satellite Observations

Liangjing Zhang[1], Henryk Dobslaw[1], Tobias Stacke[2], Andreas Güntner[1], Robert Dill[1], and Maik Thomas[1,3]

[1]Helmholtz Centre Potsdam, GFZ German Research Centre for Geosciences, 14473 Potsdam, Germany
[2]Max Planck Institute for Meteorology, Hamburg, Germany
[3]Freie Universität Berlin, Institute of Meteorology, Germany

*Correspondence to:* Liangjing Zhang (liangjing.zhang@gfz-potsdam.de)

**Abstract.** Estimates of terrestrial water storage (TWS) variations from the satellite mission GRACE are used to assess the accuracy of four global numerical model realizations that simulate the continental branch of the global water cycle. Based on four different validation metrics, we demonstrate that for the 31 largest discharge basins worldwide all model runs agree with the observations to a very limited degree only, together with large spreads among the models themselves. Since we apply a common atmospheric forcing data-set to all hydrological models considered, we conclude that those discrepancies are not entirely related to uncertainties in meteorologic input, but instead to the model structure and parametrization, and in particular to the representation of individual storage compartments with different spatial characteristics in each of the models. TWS as monitored by the GRACE mission is therefore a valuable validation data-set for global numerical simulations of the terrestrial water storage since it is sensitive to very different model physics in individual basins, which offers helpful insight to modellers for the future improvement of large-scale numerical models of the global terrestrial water cycle.

## 1 Introduction

Growing observational evidence underlines the important role of the terrestrial water cycle in shaping the Earth's climate. For instance, soil moisture variability alters the atmospheric circulation through its impact on evaporation, that affects regional and global climate (Koster et al., 2004; Meehl et al., 2009; Seneviratne and Stöckli, 2007). Snow cover raises surface albedo and isolates the land surface from the atmosphere. Groundwater also shows a significant low-frequency variability that could have regional impacts on inter-annual climate variability (Bierkens and van den Hurk, 2007). Monitoring data on water availability from both in situ and remote sensing instruments is also essential for economic and societal development. It can be used to characterize extreme hydro-meteorological conditions as flood (Chen et al., 2010) and drought (Leblanc et al., 2009). Hydrological models are important tools to forecast water resources at both short and long-term perspectives. There is now an increasing number of models that simulate the terrestrial water cycle at large spatial scales, which generally fall into two categories: Land Surface Models (LSMs) and Global Hydrology Models (GHMs). LSMs focus on solving the surface-energy balance and can be



coupled to atmospheric models, while GHMs rather focus on lateral water transfer and solving the water balance equation. Due to the different physical representation of land-surface processes, uncertainties in model structure, parameter values, and atmospheric forcing data, the performance of these models varies. There have been several model intercomparison projects, such as the Global Soil Wetness Project (GSWP; Dirmeyer et al., 2006; Dirmeyer, 2011), the Water Model Intercomparison Project

(WaterMIP; Haddeland et al., 2011), and the Inter-Sectoral Impact Model Intercomparison Project (ISI-MPI; Schewe et al., 2014) which compare the results from a multitude of models to highlight shortcomings and inconsistencies. These projects have primarily focused on evapotranspiration or soil moisture content. Gudmundsson et al. (2012) has also evaluated nine large-scale hydrological models based on runoff observations.

The terrestrial water storage (TWS), which is understood here to contain all water compartments stored above and under-

neath the land surface including soil moisture, the water content of snow-pack, land ice, surface water, and groundwater in shallow and deep aquifers, forms an important compartment of the terrestrial water cycle. It is difficult to directly measure TWS on the ground due to insufficient in-situ observations of the very diverse hydrological stores and fluxes. The terrestrial water budget method estimates TWS by solving the terrestrial water balance equation through the data of precipitation, runoff and evapotranspiration (Zeng et al., 2008; Tang et al., 2010), while TWS variations can also be derived from combined atmo-

spheric and terrestrial water-balance computations, utilizing atmospheric reanalysis data and river discharge (Seneviratne et al., 2004; Hirschi et al., 2006). However, these methods are highly dependent on the accuracy of the reanalysis data which often contain systematic errors in particular at inter-annual time scales and longer. The Gravity Recovery and Climate Experiment (GRACE) launched in 2002 provides a unique data source to estimate spatio-temporal variations of the Earth's water storage at regional up to global scales (Tapley et al., 2004; Wahr et al., 2004). Averaged over an arbitrary area with a spatial extend of

100,000 $km^2$ and greater, TWS derived from GRACE is believed to reach an accuracy of better than 1 cm equivalent water thickness (Dahle et al., 2014). Although there is a mismatch between the spatial resolution of GRACE data and that of hydrological models, the effective spatial resolution can be extrapolated to finer spatial scales through proper post-processing (Landerer and Swenson, 2012). There are now more than 13 years of GRACE data available and this length of the time-series together with a recently completed reprocessing of the whole GRACE record (Dahle et al., 2012) motivates us to revisit the

question of what can be learned from GRACE on the performance of global hydrological models in representing continental water storage variations.

Through the comparison of basin-averaged TWS from models with GRACE-based estimates, we intend to identify the advantages and deficiencies of a certain model and analyze the reasons for different model behaviors. Globally gridded TWS variations and uncertainties from GRACE estimated by the same post-processing procedure as described by Zhang et al. (2016)

are applied. We then quantitatively analyze the correspondence between TWS estimates from 4 available hydrological models and GRACE in 31 of the world's largest river basins. To separate the effects of atmospheric input data, all the models apply the same meteorological forcing data-set. Potential and actual evapotranspiration rates calculated with the different models are also analyzed. Considering the diversity of the performance of the models in these 31 basins, we focus on time series of TWS variations in two regions which are characterized by different climate regimes, i.e., the snow-dominated catchments and the

dry catchments by looking into the TWS variation time series from models and GRACE. Besides, snow, surface water and





subsurface water including root zone or/and deep layer storage from the models are also compared in order to analyze the contribution of different storage compartments to the total water storage. By investigating the relative performance of these different models, we intend to contribute to the future model development of both LSMs and GHMs.

## 2 Data set

### 2.1 Hydrological model simulations

For this study, we selected four different models to represent a broad range from conceptual hydrological to complex land surface models (Table 1). In order to ensure that this spread between the simulations is indeed related to the different representation of physics in the model, all the models are forced with the WFDEI data-set based on ERA-Interim re-analysis data (Dee et al., 2011) that has been developed during the WATCH project (Weedon et al., 2011). This WFDEI meteorological forcing dataset is a quasi-observation which combines the daily variability of the ERA-Interim re-analysis with monthly in-situ observations such as temperature and precipitation (Weedon et al., 2014). There are two precipitation products available from WFDEI: 1) corrected by using the Climate Research Unit at the University of East Anglia (CRU) observations; and (2) corrected with the Global Precipitation Climatology Centre (GPCC) data-set. Since the WFDEI data sets incorporating the CRU-based precipitation products cover a longer time span, they are used in our study and referred to subsequently as WFDEI-CRU.

The WaterGAP Global Hydrological Model (WGHM) is part of the Water-Global Assessment and Prognosis model (WaterGAP; Döll et al., 2003). WGHM is a conceptual water balance model with grossly simplified process representations. It is calibrated by tuning a runoff generation parameter against observed river discharge in a station-based calibration approach (Hunger and Döll, 2008). The model simulates the continental water cycle including the water storage compartments soil moisture within the effective root zone of vegetated areas, groundwater, canopy water, snow and surface water in rivers, lakes, reservoirs and wetlands. The latest version of WGHM as calibrated for WFDEI-GPCC forcing (version 2.2 STAN-DARD; Müller Schmied et al., 2014) is used in this study. However, we run the model with WFDEI-CRU forcing without re-calibration.

The Land Surface Discharge Model (LSDM; Dill, 2008) is based on the Simplified Land Surface Scheme (SL-Scheme) and the Hydrological Discharge Model (HD-Model; Hagemann and Gates, 2003, 2001) from the Max-Planck-Institute for Meteorology. The code has been tailored to enable the simulation of continental water mass redistribution for geodetic applications, that include the derivation of Effective Angular Momentum Functions of the continental hydrosphere to interpret and predict changes in the Earth rotation (Dobslaw et al., 2010; Dill and Dobslaw, 2010); and of vertical crustal deformations as observed from GPS permanent stations (Dill and Dobslaw, 2013).

JSBACH (Raddatz et al., 2007; Brovkin et al., 2009) is a land surface model and forms together with ECHAM6 (Stevens et al., 2013) and MPIOM (Jungclaus et al., 2013) the current Max-Planck-Institute for Meteorology's Earth System Model (MPI-ESM). As part of the MPI-ESM, JSBACH includes interactive vegetation and a 5-layer soil hydrology scheme to provide the lower atmospheric boundary conditions over land, particularly the fluxes of energy, water and momentum. For this study, how-





ever, JSBACH was used in an offline mode without interactive coupling to the other MPI-ESM compartments, but driven by prescribed WFDEI-CRU atmospheric forcing.

Finally, the Max Planck Institute of Meteorology's Hydrology Model (MPI-HM; Stacke and Hagemann, 2012) is a global hydrological model. Its water flux computations are of similar complexity to land surface models, but it does not account for any energy fluxes. Additionally to precipitation and temperature, it requires potential evapotranspiration as input which also was derived from the WFDEI using the Penman-Montheith equation similar to the Weedon et al. (2011) study.

LSDM, WGHM and MPI-HM are provided on a $0.5°$ by $0.5°$ grid, while JSBACH has a coarse resolution, with $1.875°$ spacing in longitude and irregular spacing in latitude. The mean values and the linear trends estimated over the period Jan 2003 to Dec 2012 - i.e., the common period of GRACE observations and model results - are first removed for each grid cell. Then the TWS variations are averaged over the selected basins to obtain the basin-scale TWS. Since ice dynamics and glacier mass balance are not included in the numerical models applied in this study, water mass variations in Antarctic and Greenland are not considered throughout the reminder of this paper.

## 2.2 TWS Estimates from GRACE

The U.S.-German twin satellite mission GRACE provides since April 2002 estimates of month-to-month changes in the gravitational field of the Earth mainly based on precise K-band microwave measurements of the distance between two low-flying satellites (Wahr, 2009). After correcting for short-term variability due to tides in atmosphere (Biancale and Bode, 2006), solid earth (Petit and Luzum, 2010) and oceans (Savcenko and Bosch, 2012), as well as due to non-tidal variability in atmosphere and oceans (Dobslaw et al., 2013) from the observations, the resulting gravity changes mainly represent mass transport phenomena in the Earth system, which are - apart from long-term trends - almost exclusively related to the global water cycle.

We use the monthly GRACE release 05a Level-2 products from GFZ Potsdam (Dahle et al., 2012), which can be downloaded from the website of the International Centre for Global Earth Models (icgem.gfz-potsdam.de/ICGEM). The GRACE products are expressed in terms of fully normalized spherical harmonic (SH) coefficients up to degree and order 90, approximately corresponding to a global resolution of $2°$ in latitude and longitude. We apply the same post-processing steps to the GRACE data as described by Zhang et al. (2016). The degree-1 coefficients are added following the method of Bergmann-Wolf et al. (2014). The non-isotropic filter DDK2 corresponding to an isotropic Gaussian filter with 680 km full width half maximum (Kusche, 2007; Kusche et al., 2009) is applied to remove correlated errors at particular higher degrees of the spherical harmonic expension. In order to account for signal attenuation and leakage caused by smoothing and filtering, local re-scaling factors are introduced. We use median re-scaling factors obtained from a small ensemble of global hydrological models. The gridded TWS anomalies are then estimated which can be averaged over arbitrary basins. Error estimates as a quadrature of measurement error, leakage error and re-scaling error are also provided to assess the signal-to-noise ratio (SNR) of GRACE for particular basins (full details are given in Zhang et al. (2016)). In case of a small signal-to-noise ratio, discrepancies between TWS from GRACE and models might also be attributed to comparatively large GRACE TWS errors.





## 3 Evaluation of TWS from model realizations with GRACE

We compare the basin-averaged TWS from GRACE with the results of four different numerical model realizations introduced above. In total 31 globally distributed basins, where the GRACE SNR is larger than 2 (see Fig.1 and Table 2) are selected for further study. We first focus on the global statistical performance of the models compared to GRACE. For these basins,

evaluation metrics as suggested by Gudmundsson et al. (2012) that focus both on seasonal signals and year-to-year variability are applied.

### 3.1 Evaluation metrics

First, relative annual amplitude differences are calculated according to

$$\Delta\mu = (\mu_M - \mu_O)/\mu_O, \tag{1}$$

where $\mu_O$ is the annual amplitude of the time series of TWS variations from GRACE, $\mu_M$ the annual GRACE amplitudes from the different model realizations (Fig. 2). Second, the timing of the annual cycle is assessed using phase differences of the annual harmonic for models and observations according to

$$\Delta\phi = (\phi_M - \phi_O). \tag{2}$$

If the value of $\Delta\phi$ is negative, it implies that the seasonal maximum is earlier in the year in the model than in GRACE (Fig. 3.

Annual amplitude and phase are calculated by least square regression as follows:

$$MIN \stackrel{!}{=} (\Delta\mathrm{TWS(t)} - (a + vt + A\sin(2\pi t/T + \phi))^T (\Delta\mathrm{TWS(t)} - (a + vt + A\sin(2\pi t/T + \phi)) \tag{3}$$

where $\Delta\mathrm{TWS}$ is the TWS anomaly time series, $a$ is the constant, $v$ is the trend, and $T$ is the period of one year. Third, the explained variances for all the model realizations are calculated:

$$R^2 = (var(\mathrm{TWS_O}) - \mathrm{var}(\mathrm{TWS_O} - \mathrm{TWS_M}))/\mathrm{var}(\mathrm{TWS_O}) \tag{4}$$

where $var$ denotes the variance operator. Fourth, we repeat the calculation of the explained variances for TWS time series from GRACE and the models with the mean seasonal variability removed.

### 3.2 Global evaluation

As shown in Fig. 2, the values of $\Delta\mu$ for WGHM and JSBACH are mostly negative. For JSBACH, these negative values mainly occur at mid to high latitudes of the Northern Hemisphere. WGHM underestimates the annual amplitude especially at

the low latitudes. Contrarily, MPI-HM has more basins with positive $\Delta\mu$. For LSDM, most $\Delta\mu$ values lie between -0.3 and 0.3, indicating on average better agreement of annual amplitude with GRACE. The phase difference varies more among the different models, but in most cases an earlier seasonal storage maximum is shown for the model runs relative to GRACE. There are more basins with phase difference values near zero for LSDM, while WGHM, JSBACH and MPI-HM show large differences with



respect to the GRACE result, especially in high latitudes of the Northern Hemisphere. (Fig. 3). LSDM explains the GRACE TWS variations relatively better than the other models at most basins (Fig. 4). Only in the Yukon, Nile, Zaire, Yangtze, Indus and the two basins at Australia, explained variances are less than 50%. Low values of explained variance also occur at the mid-latitude of the Northern Hemisphere for WGHM. JSBACH and MPI-HM perform generally better at basins in Africa but

have worse results in Siberia. When the annual signal is removed, the explained variances for TWS time series from GRACE and the models are generally less than 60% (Fig. 5), indicating the models's poor ability to capture the inter-annual variations. LSDM shows especially low explained variance values for many basins in Africa.

Fig. 6 summarizes the overall performance of each statistical metric for all the basins considered by means of box plots. The median $\Delta\mu$ for MPI-HM is almost zero where the other three values are all negative, indicating an underestimation of

the annual amplitude of TWS from LSDM, WGHM and JSBACH. As shown in Fig. 2d, MPI-HM overestimates the TWS variations at many basins, which compensate with those underestimated values and come to a almost zero median value. All the models have a median phase difference below zero, with LSDM having the smallest bias and range, and MPI-HM the largest bias. This shows that the TWS peaks of the models tend to proceed GRACE peaks, where LSDM performs best compared to other models. For the explained variance, LSDM shows the best median value, followed by WGHM, JSBACH and MPI-HM.

However, when the annual signal is removed, many outliers appear in LSDM for the explained variances, while WGHM and MPI-HM show slightly better performances.

We also present the basin-averaged TWS errors from GRACE and the RMS differences between TWS variations from GRACE and from the hydrological model runs (Table 2), where the largest and smallest differences are shown in bold and underlined separately. The basins are grouped according to the Köppen climate zones (Kottek et al., 2006), which include

Tropical climates, Dry climates, Temperate climates and Cold climates (see Fig. 1). For most of the basins, the GRACE errors are much smaller than the RMS differences, which indicates that the main contributions to the differences arise from model uncertainties. Out of the five basins in the tropical zone, three basins have largest differences between TWS variations from GRACE and models in LSDM. On the contrary, WGHM shows no largest differences in this climate zone. The smallest value, however, seems to occur randomly among the models. In the dry zone, most models have low SNR values and the smallest

RMS of the TWS differences are sometimes quite close to the GRACE TWS errors. For instance, at basins like Nile, Indus, and two Australian basins, the GRACE SNR estimates are all below 3. Thus, it is likely that the large uncertainty in GRACE TWS estimates contribute largely to the bad agreement in these basins. Still, MPI-HM and LSDM perform comparably better, showing a smaller number of largest differences and comparably more smallest differences. In the temperate zone, WGHM has most largest differences while MPI-HM has least. There is, however, no regular pattern of where the smallest difference occurs.

In the cold zone, all the smallest differences happen in LSDM, whereas the largest differences mainly occur at MPI-HM and JSBACH.

The performance of the models varies from basin to basin, even within the same climate zone, which could be due to the model structure, parametrization, and also the different water storage compartments included in TWS. In order to find reasons for the different model performance, we focus on two specific areas that are dominated by snow and arid climates in more



detail. There, we first assess actual evapotranspiration (AET) which is one of the main drivers for differences in the terrestrial water budget and subsequently look into the mean monthly time series of TWS and its individual storage compartments.

### 3.3 Actual evapotranspiration

As a part of terrestrial branch of the water cycle, actual evapotranspiration (AET) may explain part of the differences among the models in terms of storage variations. We choose four particularly affected basins and show the AET time series from all models (Fig. 7). Although some large differences of AET are present, the effects on subsequently simulated TWS are damped. Especially in humid areas, no direct impact can be found. For arid basins, however, the impact from AET is more dominant. For basins as Chari, Indus, Murray, and Niger, the time series comparison shows that the smaller (or larger) AET in wet season lead to higher (or lower) seasonal amplitude of TWS. Besides, in these dry areas, LSDM generally exhibits enhanced AET due to high temperatures and extremely low humidity which then lead to smaller TWS variations. As exemplarily demonstrated for the Niger basin, the relatively larger AET from LSDM covering the time period 2007 to 2009 are just correspondent to the comparably smaller TWS variations.

AET is calculated from the potential evapotranspiration (PET) as a function of the available amount of water. While starting with the same meteorological forcing data, PET is calculated differently by the models using various approaches. PET in the LSDM is calculated by the Thornthwait method, using only the daily temperature and a seasonal heat index that is based on monthly mean temperatures. In WGHM, PET is based on the Priestley-Taylor approach using net radiation, which in turn is computed as a function of incoming short-wave radiation, temperature and surface albedo. For MPI-HM, PET is computed in a pre-processing step based on Penman-Montheith using radiation, temperature, wind and humidity. JSBACH computes evaporation based on the energy balance by internally computing atmospheric water demand. Fig. 7 also displays time series comparison of PET from WGHM and LSDM. Some differences in PET are seen from these two models because of the different methods applied. These differences, however, are substantially modified when entering into AET due to the limitation of available water.

### 3.4 Snow-dominated catchments

As highlighted in section 3.2, models perform quite differently in high latitudes of the Northern Hemisphere (cold zone) which are generally dominated by snow. Especially JSBACH and MPI-HM show large differences of the TWS when compared with GRACE. We focus here on four basins in this area: Lena, Yenisei, Ob and Yukon, and look into the mean monthly time series of the TWS and its different compartments (Fig. 8). For LSDM and MPI-HM, subsurface water here only includes the water storage in the root zone, while for WGHM and JSBACH, both root zone and deep layer water storage are included. The performances of the models at those four basins are quite consistent with each other. LSDM and WGHM show the smallest phase differences with GRACE in terms of TWS while the other two exhibit negative phase shifts. The subsurface water variations from WGHM and LSDM have very similar pattern, with an apparent peak usually in May. The phases of the snow water time series from LSDM and WGHM are also quite close, but LSDM always has a slightly larger amplitude. Since the two use the same snow scheme (degree-day method), this is certainly related to the different model parameters or sub-



grid representation schemes. The surface water storage from these two models are sometimes different. For the Ob river, for instance, the different surface water storage also leads to the poor performance of WGHM in terms of TWS when compared with GRACE. The snow variations from LSDM and MPI-HM are almost identical with each other. However, the different subsurface and surface water simulated by MPI-HM causes a bad timing of the TWS peaks. For the Lena basin, although the

snow variations from LSDM, WGHM and MPI-HM are quite close, MPI-HM simulates almost no surface water variations which leads to a poor agreement of TWS with GRACE estimates. For JSBACH, there is already a large phase difference in the snow storage, which is mainly due to the poor capture of the phase of the snow accumulation and onset of melting. This could be caused by the different snow scheme applied by JSBACH. Yukon, however, is quite different from the other snow-dominated basins. Here, all the models underestimate the annual amplitude of TWS when compared with GRACE. Since the

basin-average TWS error from GRACE at Yukon is 1.19 cm and much smaller than the discrepancies between GRACE and the models (Table 2), it could be the case that all models fail to represent certain hydrological processes, or that our GRACE TWS errors are too optimistic here since the re-scaling errors are also estimated from a hydrological model ensemble. Besides, Seo et al. (2006) found also large TWS errors at Yukon basin and suggested that the atmosphere and ocean tidal and non-tidal de-aliasing errors might be a problem in this area. Investigating those discrepancies in full detail, however, is beyond the scope

of our present paper and we would like to assess the de-aliasing errors in a future study.

## 3.5 Dry catchments

We also focus on four catchments in the dry zone, which are characterized by annual precipitation smaller than annual potential evapotranspiration (McKnight and Hess, 2000). For the Nile and Niger basins, the subsurface water is the main contributor to the TWS changes (Fig. 9). The TWS variations from JSBACH and MPI-HM show a quite similar annual cycle when

compared to GRACE. MPI-HM generally exhibits a larger amplitude in simulated subsurface water and TWS. WGHM deviates considerably with a much smaller amplitude and a large phase shift in the subsurface water. The simulated surface water from WGHM brings TWS slightly closer to that from GRACE. LSDM, however, performs differently in these two basins. In Nile basin, although the subsurface water from LSDM is consistent with JSBACH and MPI-HM, the simulated surface water variations lead to a higher amplitude of TWS variations when compared with GRACE. In Niger, LSDM performs quite close

to WGHM but with a slightly larger amplitude. All models tend to perform poorly in terms of TWS when compared with GRACE in Indus basin. We note a comparably low SNR (2.2 cm) for the GRACE estimated TWS here, which is mainly contributed by the large leakage error at this basin (Zhang et al., 2016). Besides, Indus basin is not only subject to large-scale groundwater depletion from intensive irrigation, but also affected by snow melting and glaciers melting from Himalaya. Here, the subsurface water simulated by the models show already large discrepancies. As in other basins affected by snow dynamics,

JSBACH also fails to capture the snow variations properly. MPI-HM performs poorly in simulating the surface water with a delayed dynamics which leads to a preceded annual cycle. At Huang He basin, as the main contributor to the TWS, the subsurface water from LSDM, WGHM and JSBACH show similar annual variations as GRACE, while MPI-HM has a much larger amplitude. The surface water simulated differently by LSDM and WGHM then lead to different TWS variations.



## 4   Summary

We validate TWS variations simulated by four different global hydrological models with monthly GRACE gravity data. All the models are forced with the same WFDEI meteorological data-set to exclude the effect of meteorological forcing on the models. Four statistical metrics focusing on different aspects of model performance compared with GRACE have been applied. In addition, time series of TWS variations from GRACE and models are investigated, where different water storage compartments from models are shown as well.

At certain basins like Danube, Tocantins, Columbia, Ganges, Mekong, and Amazon, all numerical models show good agreement with GRACE. However, models still perform quite differently at many other basins, even though forced with the same meteorological data set. At Nile, Indus, Murray and Great Artesian Basin, large TWS errors and low SNR are found which suggests a major contribution from GRACE errors to the differences. A good capture of annual amplitude and phase at most basins leads to high values of explained variance in many basins for LSDM. However, serious problems are also found in the same model run in some central Africa basins, like Nile and Zaire, where TWS simulated by LSDM exhibits unusual large inter-annual variations. WGHM performs generally good at tropical and cold regions, but poorly at the temperate zone. JSBACH and MPI-HM show large discrepancies with GRACE at the basins in high latitudes of the Northern Hemisphere.

Model performance is also investigated in some snow dominated and dry catchments in more detail. The poor performance of JSBACH and MPI-HM in snow dominated regions is mainly related with the negative phase shifts compared to GRACE. Although MPI-HM simulates identical snow variations as LSDM, the different simulations of subsurface water and especially surface water still lead to different TWS variations in snow dominated regions. For JSBACH, the simulated snow variations show smaller amplitude and negative phase differences compared with all the other models, which also lead to the different performance of TWS. The impact from AET on TWS is relatively strong in arid areas. For instance, at some dry basins in Africa, the smaller AET from JSBACH and MPI-HM compared with LSDM and WGHM also lead to better agreement with GRACE than LSDM. At Yukon basin, we found the bad performance of all models in terms of TWS when compared with GRACE, which could be due to the effects of atmospheric and oceanic de-aliasing errors not further discussed in our current study. In future, we would like to assess all possible errors of GRACE TWS through investigation of simulated GRACE-type gravity field time-series (Flechtner et al., 2016) based on realistic orbits and instrument error assumptions as well as background error assumptions out of the updated ESA Earth System Model (Dobslaw et al., 2015, 2016) in our next step, which we believe will further help to explain the discrepancy between models and GRACE.

*Acknowledgements.* This study has been supported by the German Federal Ministry of Education and Research within the FONA research program under grants 03F0654A and 01LP1151A.





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

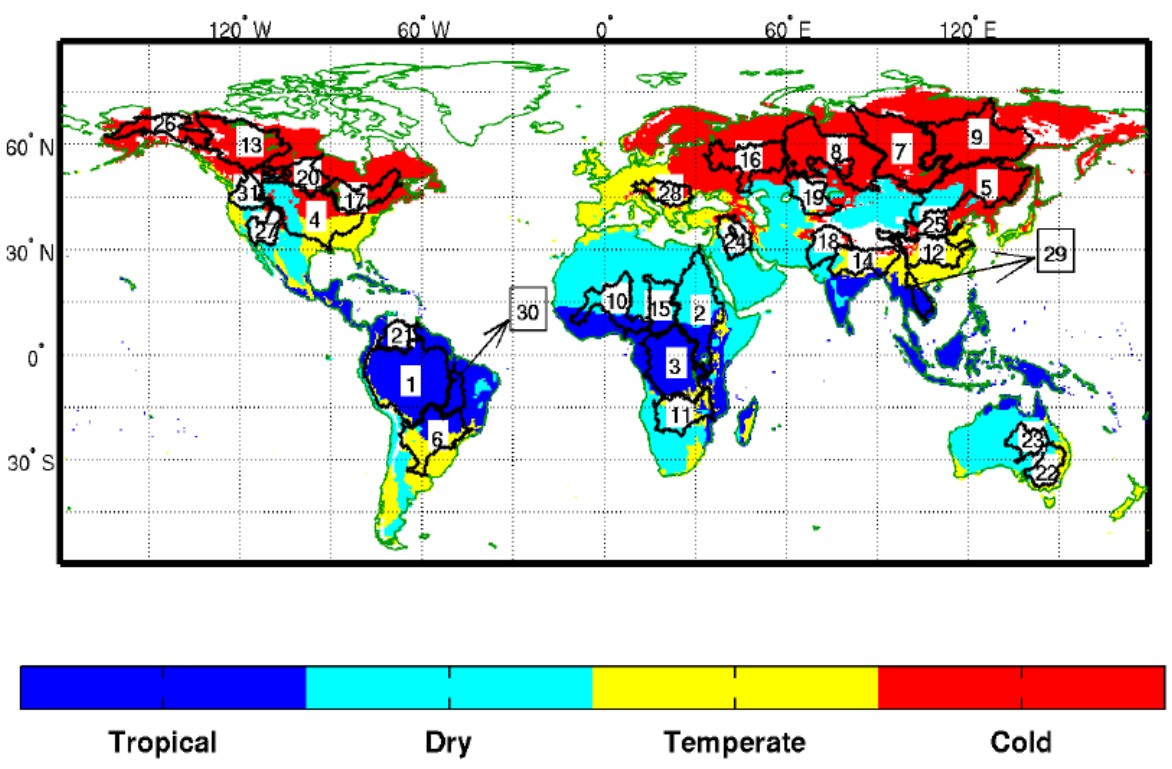

**Figure 1.** Locations of 31 globally distributed basins from the Simulated Topological Networks (STN-30p) with underlying Köppen-Geiger climate zones. Basins ID and names are indicated in Table 2.





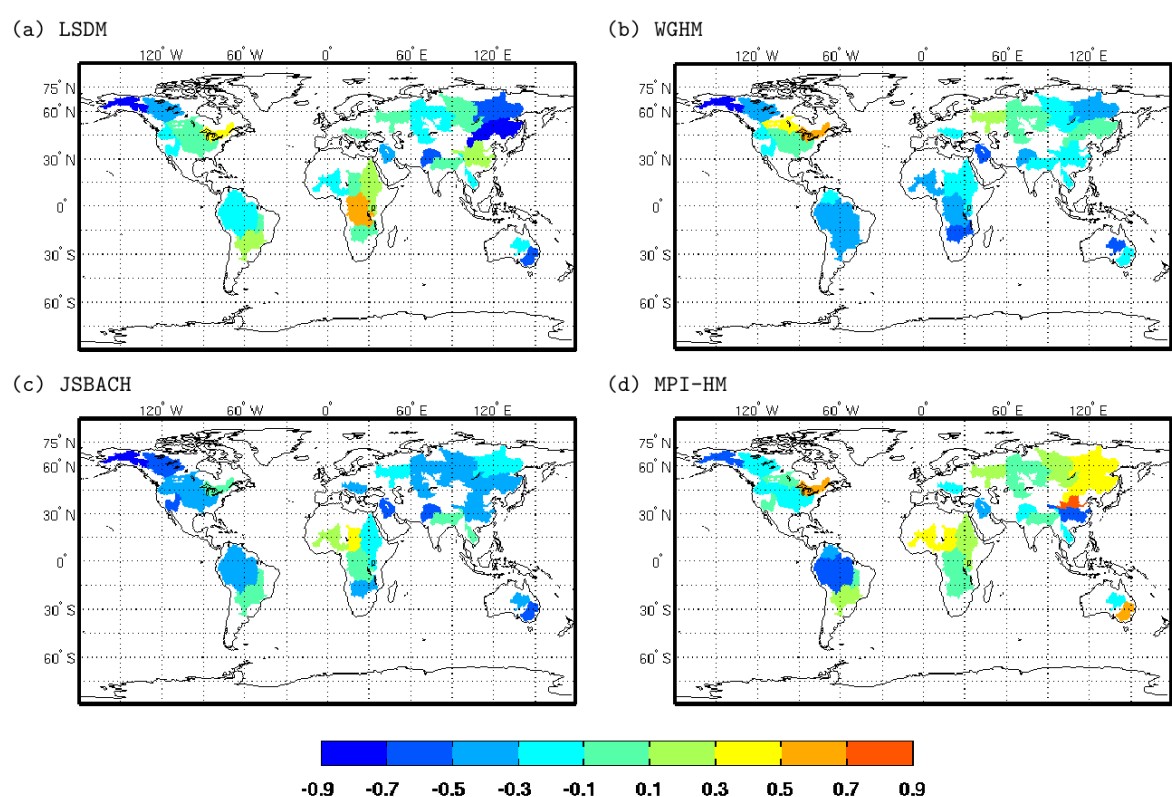

**Figure 2.** Relative amplitude differences of four hydrological model realizations with GRACE-based TWS observations.





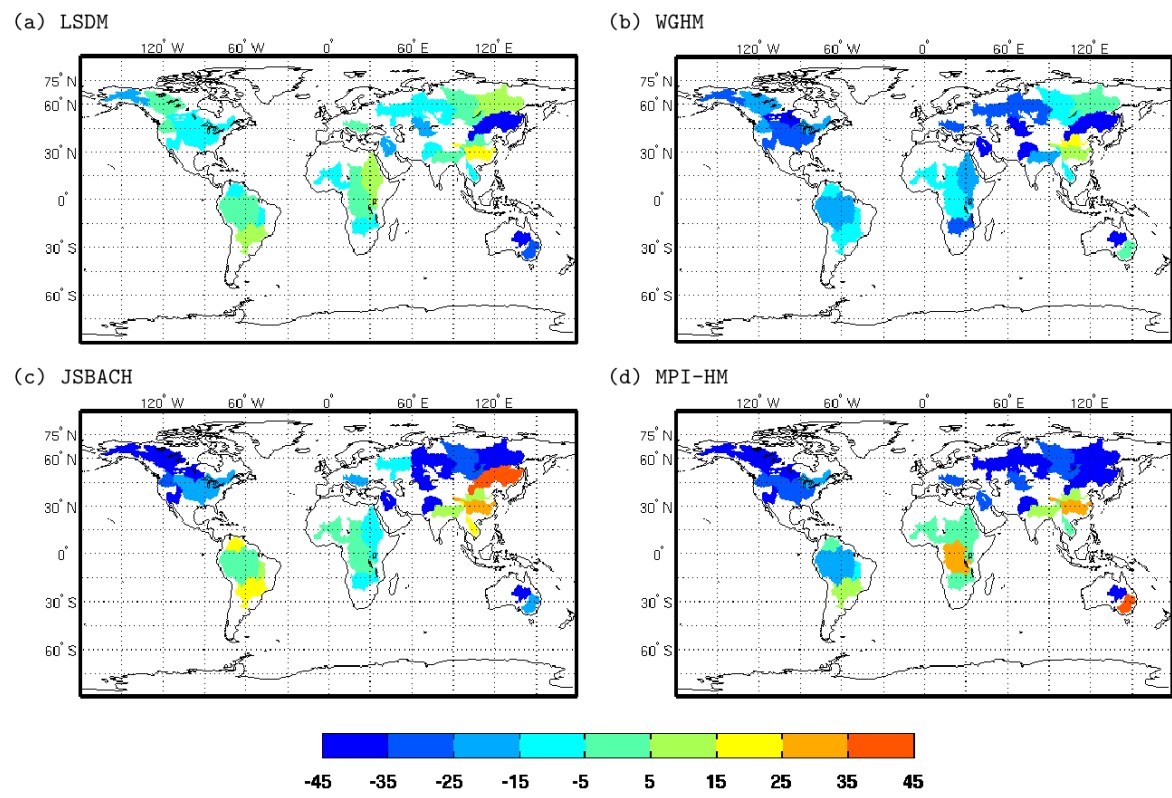

**Figure 3.** Phase differences for the annual signal of four hydrological model realizations with GRACE-based TWS observations.



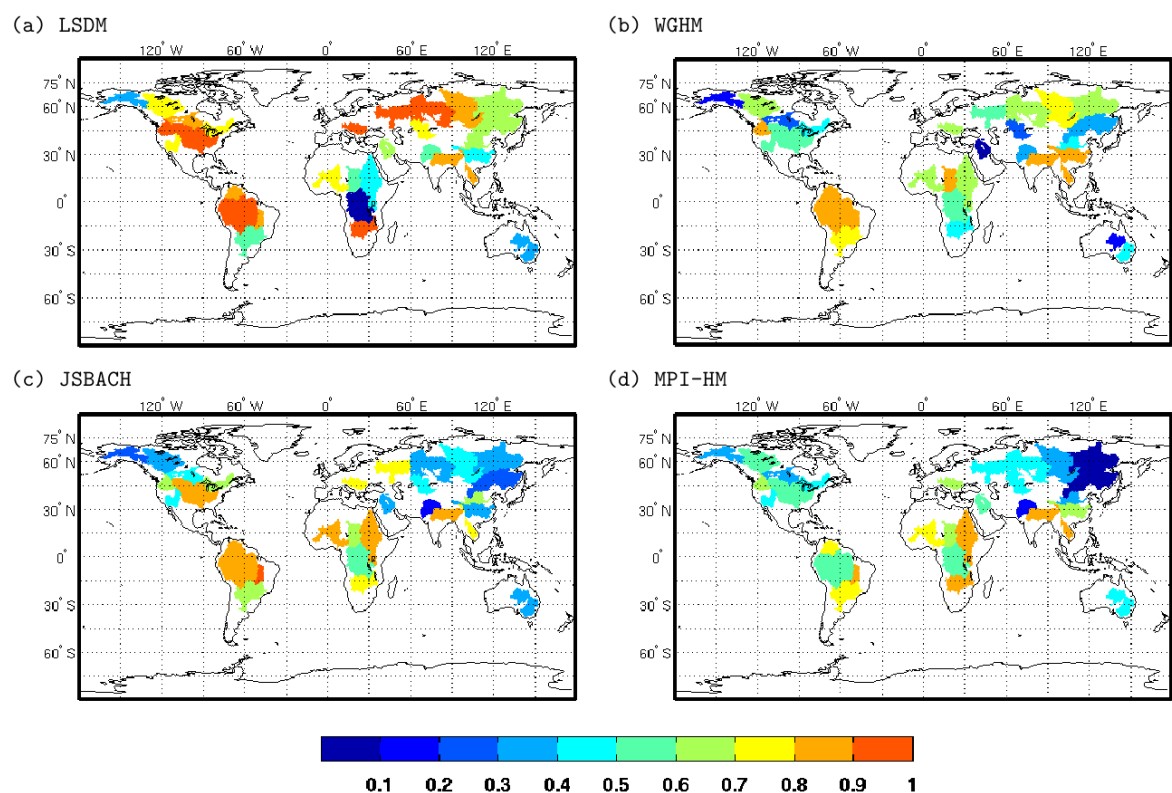

**Figure 4.** Variance of GRACE-based TWS observations that is explained by TWS as simulated in four hydrological model realizations.





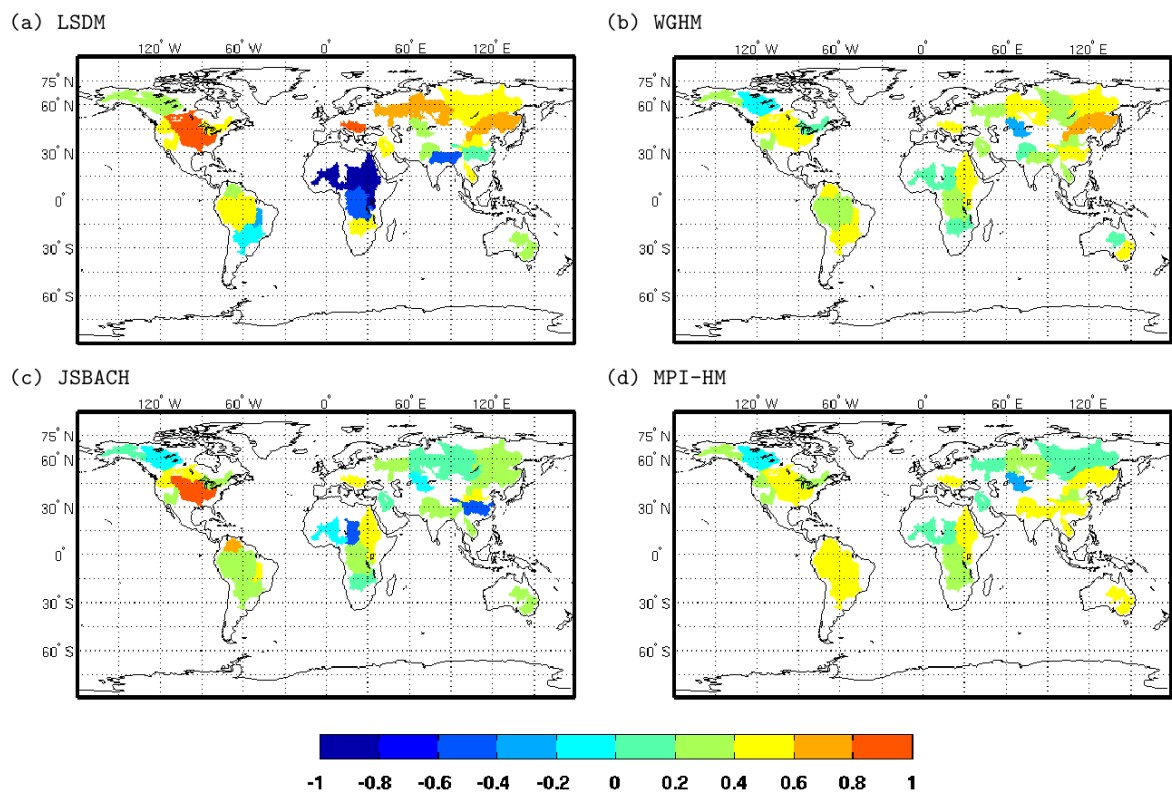

**Figure 5.** Variance of GRACE-based TWS observations that is explained by TWS as simulated in four hydrological model realizations. For both observations and model results, the annual signal has been removed.





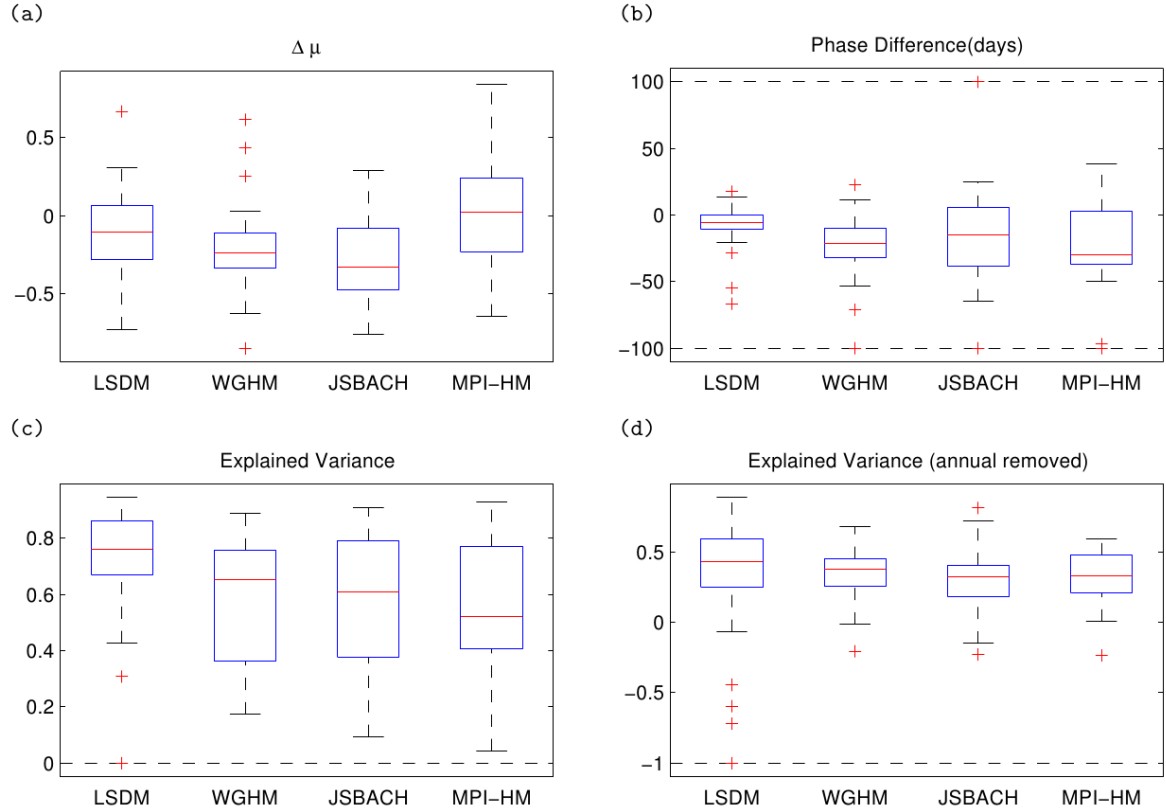

**Figure 6.** Box plots illustrating the $\Delta\mu$ (a), phase differences (b), Explained Variance (c) and Explained Variance with the annual signals removed (d) for the TWS from GRACE and models. The red horizontal line is the median, the edges of the box are the 25th and 75th percentiles, the whiskers extend to the most extreme data points not considered outliers, and outliers are plotted individually and set within the extreme data limits as indicated by the dashed line.







**Figure 7.** Monthly time series of TWS (first column) from GRACE and models, model simulated AET time series (second column) and PET time series from LSDM and WGHM (third column); each for four different catchments: Amazon, Zaire, Mekong, Niger.





**Figure 8.** Mean monthly time series of TWS (first column) and the individual storage contributions from soil moisture (second column), snow water content (third column) and surface water (fourth column); each for five snowy catchments: Ob, Lena, Yenizei and Yukon. TWS from GRACE (dashed line) has been included into every sub-figure for reference.






**Figure 9.** Mean monthly time series of TWS (first column) and the individual storage contributions from soil moisture (second column), snow water content (third column) and surface water (fourth column); each for five dry catchments: Nile, Niger, Indus and Huang He. TWS from GRACE (black line) has been included into every sub-figure for reference.



**Table 1.** Overview of the main characteristics of the four numerical models particularly considered in this study.

| Model name | Model type | Meteorological forcing variables | Storage compartments included | Soil moisture depth | Snow | Potential Evapotranspiration |
|---|---|---|---|---|---|---|
| LSDM | LSM | Precipitation, temperature | subsurface water (root zone), snow, surface water | bucket scheme without a depth | degree day | Thornthwaite |
| WGHM | GHM | Precipitation, temperature, shortwave incoming radiation | subsurface water (root zone+groundwater), snow, surface water | varies with rooting depth of land cover | degree day | Priestley-Taylor |
| JSBACH | LSM | Precipitation, temperature, wind, shortwave and longwave radiation, surface qair | subsurface water (root zone+deep layer), snow | down bedrock but at most 10 m | energy balance | physical parametrization |
| MPI-HM | GHM | Precipitation, temperature, wind, radiation, humility | subsurface water (root zone), snow, surface water | bucket scheme without a depth | degree day | Penman-Montheith |





**Table 2.** Characteristics of the basins shown in Fig 1. Bold and underlined numbers are the largest and smallest RMS differences between GRACE and models separately.

| Climate Zones | Basin ID | Name | Area (1000$km^2$) | RMSE(cm) between TWS from GRACE and | | | | GRACE TWS error(cm) | SNR |
|---|---|---|---|---|---|---|---|---|---|
| | | | | LSDM | WGHM | JSBACH | MPI-HM | | |
| Tropical | 1 | Amazon | 5853 | 4.39 | 6.08 | 5.60 | **9.53** | 1.46 | 9.76 |
| | 3 | Zaire | 3699 | **5.26** | 3.36 | 3.08 | 3.49 | 1.32 | 3.82 |
| | 21 | Orinoco | 1039 | **6.37** | 4.96 | 6.21 | 5.79 | 3.14 | 4.74 |
| | 29 | Mekong | 774 | 5.87 | 5.60 | **6.28** | 4.51 | 3.86 | 3.73 |
| | 30 | Tocantins | 769 | **7.69** | 7.49 | 4.99 | 5.45 | 2.81 | 5.95 |
| Dry | 2 | Nile | 3826 | **4.02** | 1.85 | 1.61 | 1.39 | 1.06 | 3.26 |
| | 10 | Niger | 2240 | 2.53 | **2.97** | 1.87 | 2.23 | 1.29 | 4.93 |
| | 15 | Chari | 1571 | **2.94** | 1.96 | 2.40 | 2.50 | 1.50 | 3.42 |
| | 18 | Indus | 1143 | 2.17 | 2.61 | **3.08** | 3.04 | 1.54 | 2.42 |
| | 19 | Syr-Darya | 1070 | 2.00 | **3.30** | 3.07 | 2.89 | 1.12 | 3.65 |
| | 22 | Murray | 1031 | 3.45 | 3.61 | **3.68** | 3.38 | 1.88 | 2.73 |
| | 23 | Great Artesian | 977 | 2.44 | **2.67** | 2.36 | 2.22 | 1.33 | 2.67 |
| | 24 | Shatt el Arab | 967 | 2.28 | 3.64 | **3.67** | 2.85 | 1.49 | 3.81 |
| | 25 | Huang He | 894 | 1.52 | 2.09 | 1.74 | **2.38** | 1.28 | 2.35 |
| | 27 | Colorado(Ari) | 807 | 1.90 | 2.59 | **2.98** | 2.91 | 1.41 | 2.78 |
| Temperate | 4 | Mississippi | 3203 | 1.68 | **3.54** | 2.36 | 3.45 | 0.86 | 6.60 |
| | 6 | Parana | 2661 | **4.17** | 3.03 | 3.59 | 2.81 | 1.32 | 4.50 |
| | 11 | Zambezi | 1989 | 2.89 | **7.05** | 4.83 | 3.30 | 1.57 | 6.80 |
| | 12 | Chang Jiang | 1794 | 2.58 | 2.05 | **3.24** | 3.12 | 1.49 | 3.09 |
| | 14 | Ganges | 1628 | 4.04 | **4.43** | 3.73 | 2.90 | 1.94 | 5.99 |
| Cold | 5 | Amur | 2903 | 1.20 | 1.73 | 1.88 | **2.05** | 0.68 | 3.18 |
| | 7 | Yenisei | 2582 | 1.89 | 2.34 | 3.44 | **3.54** | 0.68 | 6.67 |
| | 8 | Ob | 2570 | 1.50 | 3.20 | **4.35** | 4.14 | 0.68 | 8.31 |
| | 9 | Lena | 2418 | 2.33 | 2.40 | 3.40 | **3.99** | 0.68 | 6.01 |
| | 13 | Mackenzie | 1713 | 2.67 | 2.83 | **3.95** | 3.39 | 0.83 | 6.20 |
| | 16 | Volga | 1463 | 2.11 | 4.55 | 3.28 | **5.22** | 0.84 | 8.43 |
| | 17 | St.Lawrence | 1267 | 2.59 | 4.74 | 3.42 | **4.88** | 1.14 | 4.94 |
| | 20 | Nelson | 1047 | 1.67 | **3.87** | 3.19 | 3.31 | 1.12 | 3.82 |
| | 26 | Yukon | 852 | 5.06 | 5.72 | **5.74** | 5.29 | 1.19 | 7.68 |
| | 28 | Danube | 788 | 1.72 | 4.18 | 4.03 | **4.27** | 1.50 | 4.96 |
| | 31 | Columbia | 724 | 2.69 | 4.75 | **6.09** | 5.71 | 1.85 | 5.32 |