# Peer review of "Validation of Terrestrial Water Storage Variations as Simulated by Different Global Numerical Models with GRACE Satellite Observations"

_Hydrology and Earth System Sciences, 2016_

## Referee Comment (RC1) · Anonymous Referee #1 · 30 Aug 2016

This manuscript presents results of comparing estimated terrestrial water storage (TWS) from four hydrological models with GRACE derived TWS in 31 hydrological basins. Four metrics were used in evaluating model performances. Components of TWS as well as actual and potential ET were examined in selected basins to show the impact of model physics on estimated TWS.

The results and discussions are generally well presented and justified. But I think the paper can be further improved in a few areas. For instance, the fact that three of the four models do not model groundwater, which may contribute significantly to TWS changes, is not explicitly mentioned and discussed in the paper. In addition, the four metrics used in evaluation may be good for summarizing the differences but they do

not necessarily reflect the actual discrepancies between modeled and GRACE derived TWS. For instance, the amplitude and phase differences may not be important if TWS exhibits strong inter-annual variability.

Additional comments: Page 3, data set. Please emphasize the fact that three of the four models do not simulate groundwater and discuss its potential impacts on model estimated TWS in the result section. Also, do these models account for anthropogenic impacts such as groundwater abstraction? If not, how would this affect the comparison with TWS from GRACE which does detect changes associated with groundwater withdrawals?

Page 4, Line 8: I understand why you removed the trend but the ability to predict trend is also an important part of the models. Can you provide a scatter plot comparing trends from the models and those from GRACE in the 31 basins?

Page 5 Line 10, should the second "GRACE" be TWS?

Page 6. Line 20, I don't think it is appropriate to compare GRACE errors with the RMSE since the former represents instrument and post-processing errors and has nothing to do with how well models perform. In addition, basin-scale GRACE errors are smaller than the gridded errors which are spatially correlated (http://grace.jpl.nasa.gov/data/get-data/monthly-mass-grids-land/). Did you consider spatial correlation of errors (in both modeled and GRACE TWS) when calculating basin-scale RMSEs between the model and GRACE? Either way, I think it only makes sense to compare RMSEs among the models. The statistics in Table 2 shows the models generally did not performed well in the tropical climate. Why is that? Does it have something to do with runoff estimates as ET is energy limited in this type of climate? You don't necessarily need to collect in situ stream flow data, but some discussions and plots on runoff may be needed to explain this result.

Page 7. Line 5 to 12. As you pointed out that AET does not have a significant impact on TWS in humid areas, then what is the purpose of including three basins from that

climate in Fig. 7? I think including more basins from drier climate is more useful here.

Page 7. Line 13-22, I didn't learn anything from this paragraph and it can be removed. As you correctly pointed out that AET may be significantly different from PET which does not help much in explaining the result. Again, I think presenting runoff estimates is more useful.

Figs. 2-5: It would be very helpful if you provide time series of TWS for a basin(s) with the largest deviation from GRACE in either of these metrics. For instance, it's hard to visualize how significant a 45 degree difference in phase is. In addition, two of these metrics measure differences in seasonality which may not mean much when the interannual variability of TWS is much stronger. So providing actual TWS time series along with some discussions will be helpful for readers to understand the usefulness and limitation of these metrics.

Fig. 7. I think including runoff instead of PET is more appropriate here. Also, please try to use the same y-axis range for all plots which makes it to compare the magnitude of TWS and ET.

Fig. 8, "Subsurface water" should be soil moisture + groundwater storage for WGHM and soil moisture for all other models. Again, please use the same range for all y-axis if possible. In the caption, snow water content should be snow water equivalent.

―――――――――――――――――

---

## Author Comment (AC1) · 14 Sep 2016

**Author response to Referee #1 comments from August 30th, 2016**

We are thankful to Referee #1 for his valuable comments and suggestions, which will certainly improve our manuscript. In the following, the response to the individual comments is given with some new figures at the end. The original review is quoted in italics, whereas the author response is given in normal font.

*This manuscript presents results of comparing estimated terrestrial water storage*

[Figure]

*(TWS) from four hydrological models with GRACE derived TWS in 31 hydrological basins. Four metrics were used in evaluating model performances. Components of TWS as well as actual and potential ET were examined in selected basins to show the impact of model physics on estimated TWS.*

*The results and discussions are generally well presented and justified. But I think the paper can be further improved in a few areas. For instance, the fact that three of the four models do not model groundwater, which may contribute significantly to TWS changes, is not explicitly mentioned and discussed in the paper. In addition, the four metrics used in evaluation may be good for summarizing the differences but they do not necessarily reflect the actual discrepancies between modeled and GRACE derived TWS. For instance, the amplitude and phase differences may not be important if TWS exhibits strong inter-annual variability.*

-Except the annual amplitude and phase differences, we also show the explained variance with the seasonality removed to evaluate the agreement of models with GRACE in terms of inter-annual variability.

*Additional comments: Page 3, data set. Please emphasize the fact that three of the four models do not simulate groundwater and discuss its potential impacts on model estimated TWS in the result section. Also, do these models account for anthropogenic impacts such as groundwater abstraction? If not, how would this affect the comparison with TWS from GRACE which does detect changes associated with groundwater withdrawals?*

-Thanks for the suggestion, and this is indeed an important information. In line 7 on page 3 of the paper we propose to include the following sentences: Some of the main characteristics of the four numerical models are presented in Table 1, which provide more information on how models are different with each other. For instance, although soil moisture and snow water are included in all models, surface water and groundwater are simulated differently. JSBACH is the only model which does not include surface water. Groundwater is simulated by WGHM, where the anthropogenic impact such as groundwater abstraction is also considered. JSBACH does not include groundwater explicitly. However, soil moisture in deep layers below the root zone is simulated and buffers extreme soil moisture conditions in the layers above. Thus, some of the characteristics of real groundwater are considered. We use the term subsurface water for both soil moisture and groundwater. But the impact from consideration of groundwater to TWS variations in WGHM will be investigated in the following discussion.

A Fig. 1 will be added in the manuscript showing the differences of explained variances from WGHM with and without groundwater. The positive values indicate that WGHM with groundwater exhibits better agreement with GRACE than the one without groundwater. The large impact mainly locates at basins such as Toscantins, Niger, Huang He, Mekong and Mississippi. Only in three basins: Lena, Indus and Yukon, the differences are negative.

*Page 4, Line 8: I understand why you removed the trend but the ability to predict trend is also an important part of the models. Can you provide a scatter plot comparing trends from the models and those from GRACE in the 31 basins?*

-A scatter plot comparing trends from the models and those from GRACE in the 31 basins is shown in Fig 2, which will also be included into the final paper . The TWS trends from various models do perform quite differently among each other and with GRACE.

*Page 5 Line 10, should the second "GRACE" be TWS?*

-Yes, and it has been changed accordingly.

*Page 6. Line 20, I don't think it is appropriate to compare GRACE errors with the RMSE since the former represents instrument and post-processing errors and has nothing to do with how well models perform.*

-The GRACE errors are calculated to indicate the TWS uncertainties from GRACE, which can be applied to indicate about where GRACE might be suitable as a validation tool for models and where not. Special attention should be paid to basins with large GRACE errors, as the large discrepancies could be related to large GRACE TWS uncertainties, but not to model differences. We would like to stress that we do not directly compare GRACE errors with RMSE, but that we use the GRACE errors only as indicator of observation uncertainty. The way to estimate GRACE errors is introduced in Zhang et al. (2016). The error estimation is also investigated through an end-to-end simulation performed by Flechtner et al. (2016). We thus believe that the errors we calculated are plausible.

*In addition, basin-scale GRACE errors are smaller than the gridded errors which are spatially correlated. Did you consider spatial correlation of errors (in both modeled and GRACE TWS) when calculating basin-scale RMSEs between the model and GRACE? Either way, I think it only makes sense to compare RMSEs among the models.*

-The correlation between the gridded errors from GRACE is much larger than the one from models and is considered by using the squared exponential covariance function to estimate the statistical covariance between two grids as proposed by Landerer and Swenson (2012). The error estimates from the gridded data set also show consistent results with the ones derived directly from Stokes coefficients.

*The statistics in Table 2 shows the models generally did not performed well in the tropical climate. Why is that? Does it have something to do with runoff estimates as ET is energy limited in this type of climate? You don't necessarily need to collect in situ*

*stream flow data, but some discussions and plots on runoff may be needed to explain this result.*

-The large RMSE values in tropical regions are partly related to the fact that the TWS variability in this region is comparably large. Besides, the runoff comparison is shown for three basins affected by the tropical climate. It is seen that the bad performance of a certain model is connected with its differently simulated runoff. At Amazon basin, the positive runoff simulated from MPI-HM also leads to comparably small variability in TWS. At Zaire basin, the large inter-annual variations in TWS from LSDM are just corresponding to its runoff in an opposite way. At Mekong basin, the much larger amplitude in TWS from JSBACH compared with GRACE is related to the apparent large amplitude in its runoff.

*Page 7. Line 5 to 12. As you pointed out that AET does not have a significant impact on TWS in humid areas, then what is the purpose of including three basins from that climate in Fig. 7? I think including more basins from drier climate is more useful here.*

-This is true. However, as we add the figures of runoff comparison, three basins in tropical zone and three basins in dry climate are chosen.

*Page 7. Line 13-22, I didn't learn anything from this paragraph and it can be removed. As you correctly pointed out that AET may be significantly different from PET which does not help much in explaining the result. Again, I think presenting runoff estimates is more useful.*

-The paragraph will be shortened and we will replace the PET figures with runoff comparison.

*Figs. 2-5: It would be very helpful if you provide time series of TWS for a basin(s)*

*with the largest deviation from GRACE in either of these metrics. For instance, it's hard to visualize how significant a 45 degree difference in phase is. In addition, two of these metrics measure differences in seasonality which may not mean much when the interannual variability of TWS is much stronger. So providing actual TWS time series along with some discussions will be helpful for readers to understand the usefulness and limitation of these metrics.*

-Fig. 4 showing the time series of TWS for basins with the largest deviation from GRACE will be added in the manuscript along with some discussions.

Some sentences will be added: As each metric usually focuses only on one specific property of statistical performance and has its own limitations, the time series of TWS are shown for some basins with the largest deviation between GRACE and the model. The TWS time series are shown for Yukon basin, where both WGHM and JSBACH exhibit the largest deviation of annual amplitudes from GRACE. Although the annual amplitude is simulated better by LSDM and MPI-HM, apparent negative phase differences are shown. Amur basin is also shown, as LSDM, WGHM and MPI-HM all have the largest negative phase differences with GRACE here. Models generally capture the inter-annual signals but perform quite differently among each other and with GRACE in terms of seasonality. Almost opposite phase differences are shown for these models. The smallest explained variance for MPI-HM happens at St. Lawrence basin, where a much larger amplitude and a negative phase difference compared with GRACE are shown. When the annual signal is removed, models perform differently in terms of the explained variance. In Nile basin, large inter-annual variations sim-ulated by LSDM lead to even negative explained variance compared with other models.

*Fig. 7. I think including runoff instead of PET is more appropriate here. Also, please try to use the same y-axis range for all plots which makes it to compare the magnitude of TWS and ET.*

-The y-axis range will be changed and the runoff comparion is also shown (Fig. 3).

*Fig. 8, "Subsurface water" should be soil moisture + groundwater storage for WGHM and soil moisture for all other models. Again, please use the same range for all y-axis if possible. In the caption, snow water content should be snow water equivalent.*

-This will be changed accordingly.

**References**

Flechtner, F., Neumayer, K.-H., Dahle, C., Dobslaw, H., Fagiolini, E., Raimondo, J.-C., and Güntner, A.: What Can be Expected from the GRACE-FO Laser Ranging Interferometer for Earth Science Applications?, Surveys in Geophysics, 37, 453–470, doi:10.1007/s10712-015-9338-y, http://dx.doi.org/10.1007/s10712-015-9338-y, 2016.

Landerer, F. W. and Swenson, S. C.: Accuracy of scaled GRACE terrestrial water storage estimates, Water Resour. Res., 48, doi:10.1029/2011WR011453, W04531, 2012.

Zhang, L., Dobslaw, H., and Thomas, M.: Globally gridded terrestrial water storage variations from GRACE satellite gravimetry for hydrometeorological applications, Geophys J Int., 206, 368–378, doi:10.1093/gji/ggw153, 2016.

Please also note the supplement to this comment:
http://www.hydrol-earth-syst-sci-discuss.net/hess-2016-330/hess-2016-330-AC1-supplement.pdf

[Figure]

**Explained Variance (With - Without groundwater)**

![World map showing the differences between explained variance values from WGHM with and without groundwater, with a color scale from 0 to 0.2]

**Fig. 1.** The differences between the explained variance values from WGHM with and without groundwater.

[Figure]

**Fig. 2.** The scatter plot comparing trends from the models and those from GRACE in the 31 basins. Symbol size varies with rive basin area.

[Figure]

**Fig. 3.** Time series of TWS from GRACE and models, model simulated AET time series and model simulated runoff time series; each for six different catchments: Amazon, Zaire, Niger, Chari, Indus and Mekong.

**(a) Yukon**

**(b) Amur**

**(c) St. Lawrence**

**(d) Nile**

LSDM — WGHM — JSBACH — MPI–HM – – – GRACE

**Fig. 4.** Examples of monthly TWS time series from GRACE and models for the basins with the largest deviation between model and GRACE in each of the four metrics.

---

## Referee Comment (RC2) · Anonymous Referee #2 · 25 Sep 2016

The manuscript gives an excellent overview on the current performance of four global hydrological models validated by using the latest GRACE Release. 31 river basins worldwide covering different climate zones are assessed and deeper insights into a few basins in arid and snow dominated zones are given. The study addresses current scientific issues and goes further than previous works. It is well structures and the results are presented in a clear and comprehensible way. I think that the study will contribute to the improvement of hydrological models.

In one point the paper could still be improved: especially in Section 3.2 and in Section 4 I miss some interpretation of the findings. Is it possible to discuss a few reasons for the different model behaviors?

I have a few minor comments and questions which do not include the issues already discussed by the first reviewer.

Specific comments

Page 1, Line 7: What is the meaning of 'different spatial characteristics' of the individual storage compartments? Do you mean that e.g. groundwater is simulated using a different number of layers?

Page 2, Line 20: I think that an accuracy of 1 cm equivalent water height is a very optimistic estimate for areas as small as 100,000 kmˆ2.

Page 2, Line 24: You say, that there are more than 13 years of GRACE data available, but Figure 7 indicates that you use only 10 years of data instead of the full record. Why do you not use the whole time series of GRACE data? Is the data from the models missing?

Page 3, Data sets: For WGHM you explicitly mention the water storage compartments. Please add information about the storage compartments for the other models (to some extend you already did this in your response to the first reviewer).

Page 4, Line 27: Are the local re-scaling factors introduced for each grid cell?

Page 4, TWS Estimates from GRACE: Did you also remove the trend from the GRACE time series for the same period as for the models?

Page5, Evaluation metrics: The relative annual amplitude differences and the phase differences are interesting metrics for river basins with a strong annual cycle that follow approximately a sine curve. Can you discuss the meaning of these metrics for river basins where interannual signals dominate (also with respect to Fig. 2 and Fig. 3)?

Page 5, Line 16: Why did you estimate the trend? You subtracted it in the preprocessing step.

Page 5, Global evaluation: This paragraph gives an excellent overview on the current

performance of the four models on a global scale. Is it possible to add some inter-pretation of the results, i.e. can you explain the results by the model structure or the parametrization? E.g. why do the models in general have an earlier seasonal storage maximum than GRACE, why does LSDM have smaller phase differences than the other models, what is the problem of the models in basins with small explained variance,...?

Page 6, Line 24: Did you mean: 'most basins have low SNR values'?

Page 7, Line 8: Fig.7 shows Amazon, Zaire, Mekong, and Niger instead of Chari, Indus, Murray, and Niger. Is this intended?

Page 7, Line 9: I do not think that the performances of the models at those four basins are quite consistent with each other. In fact, JSBACH performs quite differently.

Page 9, Line 7-14: This is a good summary of the performance of the four models. Can you provide any reasons for the strengths and the deficiencies of the individual models?

Fig. 6: When the annual signal is removed the boxes become much smaller for all models except for LSDM. Do you have an explanation for this behavior?

Technical comments

Page 2, Line 12-16: Did you want to say that instead of using observations of precipi-tation (P), evapotranspiration (E), and runoff for closing the water budget equation, one can also use water vapor content and moisture flux convergence from atmospheric reanalysis data and river discharge? Is runoff and river discharge the same in both cases? Furthermore, P and E can also be taken from atmospheric reanalysis (Rodell et al. (2011) Estimating evapotranspiration using an observation based terrestrial water budget. Hydrological Processes, 25:4082–4092.) Maybe you would like to reformulate the sentence.

Page 9, Line 24-27: The structure of the sentence is strange. Probably you should delete either 'In future' or 'in our next step'.

---

## Author Response (AR1)

**Author response to Referee #1 comments from August 30th, 2016**

We are thankful to Referee #1 for his/her valuable comments and suggestions, which will certainly improve our manuscript. In the following, the response to the individual comments is given with some new figures at the end. The original review is quoted in italics, whereas the author response is given in normal font.

*This manuscript presents results of comparing estimated terrestrial water storage (TWS) from four hydrological models with GRACE derived TWS in 31 hydrological basins. Four metrics were used in evaluating model performances. Components of TWS as well as actual and potential ET were examined in selected basins to show the impact of model physics on estimated TWS.*
*The results and discussions are generally well presented and justified. But I think the paper can be further improved in a few areas. For instance, the fact that three of the four models do not model groundwater, which may contribute significantly to TWS changes, is not explicitly mentioned and discussed in the paper. In addition, the four metrics used in evaluation may be good for summarizing the differences but they do not necessarily reflect the actual discrepancies between modeled and GRACE derived TWS. For instance, the amplitude and phase differences may not be important if TWS exhibits strong inter-annual variability.*

-Besides annual amplitude and phase differences, we also show the explained variance with the seasonality removed to evaluate the agreement of models with GRACE in terms of inter-annual variability (Page 6, Line 1-2; Fig. 5 in the revised manuscript (RM)). We respond to the groundwater issue in the answer to the reviewer's next comment below.

*Additional comments: Page 3, data set. Please emphasize the fact that three of the four models do not simulate groundwater and discuss its potential impacts on model estimated TWS in the result section. Also, do these models account for anthropogenic impacts such as groundwater abstraction? If not, how would this affect the comparison with TWS from GRACE which does detect changes associated with groundwater withdrawals?*

-Thanks for the suggestion, and this is indeed an important information. In Line 12 on Page 4 of RM we propose to include the following sentences: Some of the main characteristics of the four numerical models are presented in Table 1, which provide more information on how models are different with each other. For instance, although soil moisture and snow water are included in all models, surface water and groundwater are simulated differently. JSBACH is the only model which does not include surface water. Groundwater is simulated by WGHM, where the anthropogenic impact such as groundwater abstraction is also considered. JSBACH does not include groundwater explicitly. However, soil moisture in deep layers below the root zone is simulated and buffers extreme soil moisture conditions in the layers above. Thus, some of the characteristics of real groundwater are considered. We use the term subsurface water for both soil moisture and groundwater. But the impact from consideration of groundwater to TWS variations in WGHM will be investigated in the following discussion.

A Fig. 1 is added in the manuscript showing the differences of explained variances from WGHM with and without groundwater. The positive values indicate that WGHM with groundwater exhibits better agreement with GRACE than the one without groundwater. The large impact mainly locates at basins such as Toscantins, Niger, Huang He, Mekong and Mississippi. Only in three basins (Lena, Indus and Yukon), the effect of groundwater consideration in the model is negative.

*Page 4, Line 8: I understand why you removed the trend but the ability to predict trend is also an important part of the models. Can you provide a scatter plot comparing trends from the models and those from GRACE in the 31 basins?*

-A scatter plot comparing trends from the models and those from GRACE in the 31 basins is shown in Fig 2. The TWS trends from various models do perform quite differently among each other and with GRACE.

*Page 5 Line 10, should the second "GRACE" be TWS?*

-Yes, and it has been changed accordingly.

*Page 6. Line 20, I don't think it is appropriate to compare GRACE errors with the RMSE since the former represents instrument and post-processing errors and has nothing to do with how well models perform.*

-The GRACE errors are calculated to indicate the TWS uncertainties from GRACE, which can be applied to indicate about where GRACE might be suitable as a validation tool for models and where not. Special attention should be paid to basins with

large GRACE errors, as the large discrepancies could be related to large GRACE TWS uncertainties, but not to model differences. We would like to stress that we do not directly compare GRACE errors with RMSE, but that we use the GRACE errors only as indicator of observation uncertainty. The way to estimate GRACE errors is introduced in Zhang et al. (2016). The error estimation is also investigated through an end-to-end simulation performed by Flechtner et al. (2016). We thus believe that the
5 errors we calculated are plausible.

*In addition, basin-scale GRACE errors are smaller than the gridded errors which are spatially correlated. Did you consider spatial correlation of errors (in both modeled and GRACE TWS) when calculating basin-scale RMSEs between the model and GRACE? Either way, I think it only makes sense to compare RMSEs among the models.*
10    -The correlation between the gridded errors from GRACE is much larger than the one from models and is considered by using the squared exponential covariance function to estimate the statistical covariance between two grids as proposed by Landerer and Swenson (2012). The error estimates from the gridded data set also show consistent results with the ones derived directly from Stokes coefficients.

15 *The statistics in Table 2 shows the models generally did not performed well in the tropical climate. Why is that? Does it have something to do with runoff estimates as ET is energy limited in this type of climate? You don't necessarily need to collect in situ stream flow data, but some discussions and plots on runoff may be needed to explain this result.*
    -The large RMSE values in tropical regions are partly related to the fact that the TWS variability in this region is comparably large. Besides, the runoff comparison is shown for three basins affected by the tropical climate (Fig. 3). The runoff is
20 calculated from the models following the equation: Runoff=Precipitation-Actual evapotranspiration-TWSC (Ramillien et al., 2006). It is seen that the performance of a certain model is connected with its differently simulated runoff. At Amazon basin, the comparably large runoff simulated from MPI-HM also leads to smaller variability in TWS, which is also shown at Orinoco basin. At Mekong basin, the larger amplitude in TWS from JSBACH compared with GRACE is related to the apparent small amplitude in its runoff.

*Page 7. Line 5 to 12. As you pointed out that AET does not have a significant impact on TWS in humid areas, then what is the purpose of including three basins from that climate in Fig. 7? I think including more basins from drier climate is more useful here.*
    -This is true. We remove the basins in humid areas and focus on three basins (Niger, Chari and Indus) in the dry zone in Fig.
30 9 (RM).

*Page 7. Line 13-22, I didn't learn anything from this paragraph and it can be removed. As you correctly pointed out that AET may be significantly different from PET which does not help much in explaining the result. Again, I think presenting runoff estimates is more useful.*
    -Following the reviewer's suggestion we shorten the discussion on PET in the revised manuscript and add a figure showing
35 comparison of runoff data (Fig. 10 (RM)).

*Figs. 2-5: It would be very helpful if you provide time series of TWS for a basin(s) with the largest deviation from GRACE in either of these metrics. For instance, it's hard to visualize how significant a 45 degree difference in phase is. In addition,*
40 *two of these metrics measure differences in seasonality which may not mean much when the interannual variability of TWS is much stronger. So providing actual TWS time series along with some discussions will be helpful for readers to understand the usefulness and limitation of these metrics.*
    -Fig. 4 showing the time series of TWS for basins with the largest deviation from GRACE is added in the manuscript along with some discussions.
45    In Line 22 of Page 6 (RM), some sentences are added: As each metric usually focuses only on one specific property of statistical performance and has its own limitations, the time series of TWS are given for some basins with the largest deviation between GRACE and the model. We choose Yukon basin, where both WGHM and JSBACH exhibit the largest deviation of annual amplitudes from GRACE. Although the annual amplitude is simulated better by LSDM and MPI-HM, apparent negative phase differences are shown. Amur basin is also shown, as LSDM, WGHM and MPI-HM all have the largest negative phase

differences with GRACE here. Models generally capture the inter-annual signals but perform quite differently among each other and with GRACE in terms of seasonality. Almost opposite phase differences are found for these models. The smallest explained variance for MPI-HM happens at St. Lawrence basin, where a much larger amplitude and a negative phase difference compared with GRACE are found. When the annual signal is removed, models perform differently in terms of the explained variance. In Nile basin, large inter-annual variations simulated by LSDM lead to even negative explained variance compared with the other models.

*Fig. 7. I think including runoff instead of PET is more appropriate here. Also, please try to use the same y-axis range for all plots which makes it to compare the magnitude of TWS and ET.*
   -The y-axis range is changed and the runoff comparison is also shown (Fig. 3).

*Fig. 8, "Subsurface water" should be soil moisture + groundwater storage for WGHM and soil moisture for all other models. Again, please use the same range for all y-axis if possible. In the caption, snow water content should be snow water equivalent.*
   -This has been changed accordingly.

**References**

Flechtner, F., Neumayer, K.-H., Dahle, C., Dobslaw, H., Fagiolini, E., Raimondo, J.-C., and Güntner, A.: What Can be Expected from the GRACE-FO Laser Ranging Interferometer for Earth Science Applications?, Surveys in Geophysics, 37, 453–470, doi:10.1007/s10712-015-9338-y, http://dx.doi.org/10.1007/s10712-015-9338-y, 2016.

5  Landerer, F. W. and Swenson, S. C.: Accuracy of scaled GRACE terrestrial water storage estimates, Water Resour. Res., 48, doi:10.1029/2011WR011453, W04531, 2012.

Ramillien, G., Frappart, F., Güntner, A., Ngo-Duc, T., Cazenave, A., and Laval, K.: Time variations of the regional evapotranspiration rate from Gravity Recovery and Climate Experiment (GRACE) satellite gravimetry, Water Resources Research, 42, n/a–n/a, doi:10.1029/2005WR004331, http://dx.doi.org/10.1029/2005WR004331, w10403, 2006.

10 Zhang, L., Dobslaw, H., and Thomas, M.: Globally gridded terrestrial water storage variations from GRACE satellite gravimetry for hydrometeorological applications, Geophys J Int., 206, 368–378, doi:10.1093/gji/ggw153, 2016.

Explained Variance (With - Without groundwater)

[Figure]

**Figure 1.** The differences between the explained variance values from WGHM with and without groundwater.

[Figure]

**Figure 2.** The scatter plot comparing trends from the models and those from GRACE in the 31 basins. Symbol size varies with rive basin area.

[Figure]

**Figure 3.** Time series of TWS (left) from GRACE and models and model simulated runoff time series (right); each for three different catchments in tropical zone: Amazon, Orinoco, and Mekong.

[Figure]

**Figure 4.** Examples of monthly TWS time series from GRACE and models for the basins with the largest deviation between model and GRACE in each of the four metrics: Relative amplitude differences (Yukon), phase differences (Amur), explained variance (St. Lawrence) and explained variance with annual signal removed (Nile).

**Author response to Referee #2 comments from September 25th, 2016**

We are thankful to Referee #2 for the constructive comments and suggestions, which will certainly improve our manuscript. In the following, the response to the individual comments is given. The original review is quoted in italics, whereas the author response is given in normal font.

*The manuscript gives an excellent overview on the current performance of four global hydrological models validated by using the latest GRACE Release. 31 river basins worldwide covering different climate zones are assessed and deeper insights into a few basins in arid and snow dominated zones are given. The study addresses current scientific issues and goes further than previous works. It is well structures and the results are presented in a clear and comprehensible way. I think that the study will contribute to the improvement of hydrological models. In one point the paper could still be improved: especially in Section 3.2 and in Section 4 I miss some interpretation of the findings. Is it possible to discuss a few reasons for the different model behaviors? I have a few minor comments and questions which do not include the issues already discussed by the first reviewer.*

-We would like to thank the referee for the positive comments on the manuscript. We add some discussions on the reasons for the different model behaviors in Line 9 of Page 10 (RM), which is further explained in the response to the specific comments in the following.

*Page 1, Line 7: What is the meaning of 'different spatial characteristics' of the individual storage compartments? Do you mean that e.g. groundwater is simulated using a different number of layers?*

-By 'different spatial characteristics', we mean that the spatial auto-correlation pattern of water storage is different for the different storage compartments. For instance, surface water exhibits a linear or point-like pattern as it is concentrated in areas such as rivers and lakes, whereas soil moisture and groundwater tend to have a smoother distribution in space with larger spatial correlation lengths. For hydrological models, a missing storage compartment, such as surface water in JSBACH, thus leads to a different spatial pattern of TWS variability compared to the other three models.

*Page 2, Line 20: I think that an accuracy of 1 cm equivalent water height is a very optimistic estimate for areas as small as 100,000 km^2.*

-Indeed, with this sentence we intend to indicate the limit of what could be achieved with GRACE, according to the cited reference.

*Page 2, Line 24: You say, that there are more than 13 years of GRACE data available, but Figure 7 indicates that you use only 10 years of data instead of the full record. Why do you not use the whole time series of GRACE data? Is the data from the models missing?*

-Yes, the 10 years chosen here is the common period of data from GRACE and the hydrological models available to us.

*Page 3, Data sets: For WGHM you explicitly mention the water storage compartments. Please add information about the storage compartments for the other models (to some extend you already did this in your response to the first reviewer).*

-The different water storage compartments simulated by the models are shown in Table 1 of the manuscript. Still, to be consistent and clear, some more sentences are added for each of the other models: The global water storage variations contain surface water in rivers, lakes and wetlands, groundwater and soil moisture, as well as water stored in snow and ice (LSDM). Snow is treated as external layers above the soil column, with maximum of five snow layers. Soil moisture in deep layers below the root zone is simulated and buffers extreme soil moisture conditions in the layers above (JSBACH). TWS from MPI-HM is simulated as the sum of soil moisture in the root zone, snow and surface water (MPI-HM).

*Page 4, Line 27: Are the local re-scaling factors introduced for each grid cell?*

-Yes, and 'for each grid cell' is added after the sentence.

*Page 4, TWS Estimates from GRACE: Did you also remove the trend from the GRACE time series for the same period as for the models?*

-Yes, and one sentence in Line 8 on Page 5 (RM) has been added to make it more clear: As for the model data, the linear trend is removed over the period Jan 2003 to Dec 2012.

*Page5, Evaluation metrics: The relative annual amplitude differences and the phase differences are interesting metrics for river basins with a strong annual cycle that follow approximately a sine curve. Can you discuss the meaning of these metrics for river basins where interannual signals dominate (also with respect to Fig. 2 and Fig. 3)?*
-The seasonal cycle and the inter-annual variations of the signal are investigated separately by different metrics in our study. The explained variance (eq. 4) is also calculated for de-seasonalized time series to evaluate model performance with respect to inter-annual variations. Fig. 2 and Fig. 3 (RM) just focused on the seasonal variations from the models where the inter-annual signals were not taken into account.

*Page 5, Line 16: Why did you estimate the trend? You subtracted it in the preprocess- ing step.*
-It is true that we removed the trend before the calculation of seasonal and inter-annual metrics to avoid any influence of the trend on them and it should be removed from the equation.

*Page 5, Global evaluation: This paragraph gives an excellent overview on the current performance of the four models on a global scale. Is it possible to add some inter- pretation of the results, i.e. can you explain the results by the model structure or the parametrization? E.g. why do the models in general have an earlier seasonal storage maximum than GRACE, why does LSDM have smaller phase differences than the other models, what is the problem of the models in basins with small explained variance,. . .?*
-As models perform differently in different areas and are affected by combined impacts from their parametrization, structure, and physical representation, it is hard to interpret the general global performance and to single out specific reasons for poor or bad model performance with respect to structure and parametrization. For instance, it is assumed that missing water storage compartments in a certain model is the main reason for the earlier seasonal storage maximum than GRACE in the previous work. Through our investigation, it is found that this is not totally true. Groundwater is missing in LSDM, but still it shows smaller phase differences than the other models. Besides, the negative phase difference in JSBACH is found to be more related to its snow water simulation. We discuss these and other related issues on model performance for specific regions in the manuscript, and also consider AET and runoff (besides TWS time series) for an interpretation.

*Page 6, Line 24: Did you mean: 'most basins have low SNR values'?*
-Yes, it has been changed accordingly.

*Page 7, Line 8: Fig.7 shows Amazon, Zaire, Mekong, and Niger instead of Chari, Indus, Murray, and Niger. Is this intended?*
-Fig. 7 is updated to Fig. 9 (RM) with focus on three basins in dry climate (Chari, Indus, and Mekong).

*Page 7, Line 9: I do not think that the performances of the models at those four basins are quite consistent with each other. In fact, JSBACH performs quite differently.*
-This sentence is indeed misleading and has been removed.

*Page 9, Line 7-14: This is a good summary of the performance of the four models. Can you provide any reasons for the strengths and the deficiencies of the individual models?*
-As noted in a comment above, sorting out the specific reasons for strengths and deficiencies of a specific model for a certain region is difficult due to model complexity. Nevertheless, we try to work out some reasons and extend the third paragraph in the Summary (RM) as follows: Model performance is also investigated in some snow dominated and dry catchments in more detail through time series comparison. The poor performance of JSBACH and MPI-HM in snow dominated regions is mainly related to negative phase shifts compared to GRACE. MPI-HM simulates identical snow variations as LSDM, however, the different simulations of subsurface water and especially surface water still lead to different TWS variations in snow dominated regions. Despite of the missing surface water compartment, the simulated snow variations in JSBACH already show smaller amplitude and negative phase differences compared with all the other models. This could be related to the fact that JSBACH

simulates snow in a more physical way based on energy balance, which is totally different from the degree-day method applied by all the other models. The comparably better agreement of LSDM and WGHM with GRACE in terms of TWS in these snow dominated basins is partly caused by the realistic surface water compartment represented by these two models. In the dry catchments, the impact from AET on TWS is relatively strong. The smaller AET from MPI-HM also leads to better agreement with
5  GRACE, whereas LSDM shows large differences with GRACE in terms of TWS especially at some dry basins in central Africa partly due to the too simple evaporation scheme. PET is simulated using a superior parametrization by MPI-HM, while LSDM applies still the traditional Thornthwaite method based solely on air temperature. The groundwater considered by WGHM also has some impact on the simulated TWS, especially at basins as Toscantins, Mekong, Niger and Mississippi. At Yukon basin, we found the bad performance of all models in terms of TWS when compared with GRACE, which could be due to the effects
10  of atmospheric and oceanic de-aliasing errors not further discussed in our current study. In future, we would like to assess all possible errors of GRACE TWS through investigation of simulated GRACE-type gravity field time-series (Flechtner et al., 2016) based on realistic orbits and instrument error assumptions as well as background error assumptions out of the updated ESA Earth System Model (Dobslaw et al., 2015, 2016), which we believe will further help to explain the discrepancy between global models of the terrestrial water cycle and GRACE satellite observations.

*Fig. 6: When the annual signal is removed the boxes become much smaller for all models except for LSDM. Do you have an explanation for this behavior?*
    -The comparably large spread of the variations for this metric (explained variance with the annual signal removed) from LSDM is related to the fact that the performance of the inter-annual signal simulation from LSDM varies among the different
20  areas. When the annual signal is removed, the explained variance remains high at some basins in high latitudes of the Northern Hemisphere, whereas much larger inter-annual variations are simulated at central Africa which leads to quite low explained variance values. Furthermore, please note the different scale of the y-axis in both plots which may suggest a considerable reduction of the spread of the metric for the case with the annual signal removed, which actually is not that large in absolute values.

*Technical comments Page 2, Line 12-16: Did you want to say that instead of using observations of precipi- tation (P), evapotranspiration (E), and runoff for closing the water budget equation, one can also use water vapor content and moisture flux convergence from atmospheric reanalysis data and river discharge? Is runoff and river discharge the same in both cases? Furthermore, P and E can also be taken from atmospheric reanalysis (Rodell et al. (2011) Estimating evapotranspiration using an*
30  *observation based terrestrial water budget. Hydrological Processes, 25:4082–4092.) Maybe you would like to reformulate the sentence.*
    -Thanks for the suggestions. To be more clear, the sentence is changed to: The terrestrial water budget method estimates TWS by solving the terrestrial water balance equation through the data of precipitation, runoff and evapotranspiration from observations and atmospheric reanalysis (Zeng et al., 2008; Tang et al., 2010; Rodell et al. 2011), while TWS variations can also be
35  derived from combined atmospheric and terrestrial water-balance computations, utilizing water vapor content and moisture flux convergence from atmospheric reanalysis data and river discharge measurements (Seneviratne et al., 2004; Hirschi et al., 2006).

*Page 9, Line 24-27: The structure of the sentence is strange. Probably you should delete either 'In future' or 'in our next step'.*
    -It is true, and 'in our next step' is removed.

40

**References**

[revised manuscript text omitted]